# Minimally Invasive Therapies for Knee Osteoarthritis

**DOI:** 10.3390/jpm14090970

**Published:** 2024-09-13

**Authors:** Uchenna Osuala, Megan H. Goh, Arian Mansur, John B. Smirniotopoulos, Arielle Scott, Christine Vassell, Bardia Yousefi, Neil K. Jain, Alan A. Sag, Allison Lax, Kevin W. Park, Alexander Kheradi, Marc Sapoval, Jafar Golzarian, Peiman Habibollahi, Osman Ahmed, Shamar Young, Nariman Nezami

**Affiliations:** 1Georgetown University School of Medicine, Washington, DC 20007, USA; uco2@georgetown.edu (U.O.); john.b.smirniotopoulos@medstar.net (J.B.S.); 2Harvard Medical School, Cambridge, MA 02115, USA; megangoh@hms.harvard.edu (M.H.G.); arianmansur@hms.harvard.edu (A.M.); 3Division of Vascular and Interventional Radiology, MedStar Washington Hospital Center, Washington, DC 20010, USA; neil.k.jain@medstar.net; 4Department of Bioengineering, University of Maryland College Park, College Park, MD 20742, USA; ascott10@terpmail.umd.edu (A.S.); cvassell@terpmail.umd.edu (C.V.); bardia.yousefi@gmail.com (B.Y.); 5Division of Vascular and Interventional Radiology, Department of Radiology, Duke University Medical Center, Durham, NC 27705, USA; alan.sag@duke.edu; 6Department of Radiology, MedStar Georgetown University Hospital, Washington, DC 20007, USA; allison.lax@gunet.georgetown.edu; 7Department of Orthopaedic Surgery, MedStar Georgetown University Hospital, Washington, DC 20007, USA; kevin.w.park@medstar.net; 8Department of Emergency Medicine, MedStar Georgetown University Hospital, Washington, DC 20007, USA; alexander.r.kheradi@medstar.net; 9Hôpital Européen Georges-Pompidou, 75015 Paris, France; marc.sapoval@gmail.com; 10North Star Vascular and Interventional Institute, Minnesota, MN 55427, USA; jafgol@gmail.com; 11Department of Radiology, Division of Vascular and Interventional Radiology, University of Minnesota Medical School, Minneapolis, MN 55455, USA; 12Department of Interventional Radiology, The University of Texas MD Anderson Cancer Center, Houston, TX 77030, USA; phabibollahi@mdanderson.org; 13Division of Interventional Radiology, Rush University Medical Center, Chicago, IL 60612, USA; osman1423@gmail.com; 14Division of Interventional Radiology, Department of Medical Imaging, University of Arizona Medical Center, Tucson, AZ 85712, USA; shamar@radiology.arizona.edu; 15Division of Vascular and Interventional Radiology, MedStar Georgetown University Hospital, Washington, DC 20007, USA; 16Lombardi Comprehensive Cancer Center, Washington, DC 20007, USA

**Keywords:** knee osteoarthritis, genicular artery embolization, genicular nerve ablation, regenerative and experimental therapies, AI for knee osteoarthritis

## Abstract

Knee osteoarthritis (KOA) is a musculoskeletal disorder characterized by articular cartilage degeneration and chronic inflammation, affecting one in five people over 40 years old. The purpose of this study was to provide an overview of traditional and novel minimally invasive treatment options and role of artificial intelligence (AI) to streamline the diagnostic process of KOA. This literature review provides insights into the mechanisms of action, efficacy, complications, technical approaches, and recommendations to intra-articular injections (corticosteroids, hyaluronic acid, and plate rich plasma), genicular artery embolization (GAE), and genicular nerve ablation (GNA). Overall, there is mixed evidence to support the efficacy of the intra-articular injections that were covered in this study with varying degrees of supported recommendations through formal medical societies. While GAE and GNA are more novel therapeutic options, preliminary evidence supports their efficacy as a potential minimally invasive therapy for patients with moderate to severe KOA. Furthermore, there is evidentiary support for the use of AI to assist clinicians in the diagnosis and potential selection of treatment options for patients with KOA. In conclusion, there are many exciting advancements within the diagnostic and treatment space of KOA.

## 1. Introduction

Knee osteoarthritis (KOA) is a musculoskeletal disorder characterized by articular cartilage degeneration and chronic inflammation. This review presents a broad overview of the diagnosis, minimally invasive therapeutic options, and recent innovations in decision-making to optimize the treatment of KOA. This article explores the mechanisms of action, efficacy, and complications associated with intra-articular steroid, hyaluronic acid, and platelet-rich plasma injections, as well as the technique and approach to intra-articular injections. Furthermore, this study explores more recent innovative therapeutic and diagnostic options for KOA such as genicular artery embolization, genicular nerve ablation, regenerative and experimental therapies, and the role of artificial intelligence (AI) in decision-making and patient selection.

## 2. Osteoarthritis Background

### 2.1. Epidemiology, Etiology, and Risk Factors

KOA affects a significant portion of the population, with an estimated global prevalence of 22.9% in adults aged 40 and older. The etiology of KOA is multifactorial, with risk factors including advancing age, genetic predisposition, previous knee injuries, and obesity. These factors contribute to the progressive degeneration of articular cartilage, leading to the characteristic symptoms of pain and stiffness in KOA [1,2].

### 2.2. Diagnosis: Clinical Features and Image Findings

Diagnosis of KOA typically involves a combination of clinical examination and imaging. Clinicians assess the knee for signs of joint line tenderness, swelling, and decreased range of motion. Imaging is also crucial for confirming the diagnosis and assessing the extent of joint damage. Plain film radiographs are the most used imaging modality when diagnosing KOA. The Kellgren and Lawrence (KL) grading system (Table 1) is a widely accepted scheme for characterizing KOA based on radiographic features of joint space narrowing, the presence of osteophytes, subchondral sclerosis, and changes to the shape of the bone [3]. These representative changes can be seen in Figure 1. While the KL grading system is useful, radiographs are inherently limited in their ability to precisely measure the extent of the pathological processes in KOA. This is evident when there are discrepancies between radiographic findings and clinical features in some patients. One study found that the proportion of patients with radiographic KOA and pain ranged from 15% to 81%. The authors conclude that radiographs alone are an imprecise predictor of pain or disability [4].

Magnetic resonance imaging (MRI) provides a more detailed view of the knee and surrounding structures. As shown in Figure 2, cartilage quality can be directly assessed with MRI and the presence of effusion, synovitis, and bone marrow changes can be more accurately detected. These features are associated with knee pain and are helpful in alerting providers to the acuity and extent of disease [5]. MRIs are also useful in identifying concomitant pathologies such as ligamental and meniscal injuries which can alter the course of OA [6]. While the higher fidelity of MRIs allows for a more detailed understanding of knee pathology, the burden of cost and limitations of access should also be considered.

### 2.3. Current Management and Challenges in Treatments

Management strategies for KOA aim to alleviate pain, improve function, and maintain quality of life (Table 2).

Non-operative treatments include weight-loss programs, physical therapy, and pharmacologic interventions such as topical analgesics and oral nonsteroidal anti-inflammatory drugs (NSAIDs). While NSAIDs can be effective when used in conjunction with other therapies, chronic use increases the risk of gastric ulcers, renal injury, and cardiovascular toxicity, especially in patients with comorbid conditions [7]. Weight loss, especially when combined with physical therapy, has been shown to improve pain, function, and stiffness with no adverse effects. However, the therapeutic effect of these modalities is limited by access, patient compliance, and disease stage/severity [8,9,10]. Other modalities like electro-frequency, photodynamic therapy, infrared heat therapy, electro-frequency, photodynamic therapy, infrared heat therapy, and even acupuncture as traditional homeopathy are quite minimally invasive; however, their clinical efficacy is uncertain or quite limited.

Operative treatments range from arthroscopic procedures to total knee arthroplasty (TKA) for advanced cases. Minimally invasive treatments consist of intra-articular injections of corticosteroids (IAC), hyaluronic acid (HA), or platelet-rich plasma (PRP), and non-surgical image-guided procedures like genicular nerve ablation (GNA) and genicular artery embolization (GNE). These minimally invasive therapies are increasing in popularity due to superior symptom relief over non-operative management while avoiding the risks of surgery [8,9,10,11].

Despite the availability of various treatment options, managing KOA remains challenging due to its chronic nature and the variability in patient response to treatments. Some patients may experience insufficient relief from conservative measures, while others may not be suitable candidates for surgery. Additionally, the potential side-effects and complications associated with different treatments can limit their use [12].

## 3. Intra-Articular Steroid Injections

### 3.1. Background and Mechanism of Action of Intra-Articular Steroid Injections

Intra-articular corticosteroid (IAC) injections are a commonly used tool in the management of mild to moderate KOA. Corticosteroids provide relief through direct action on nuclear steroid receptors, which suppress the release of inflammatory mediators, leading to decreased synovitis and pain relief [13]. IAC injections are typically administered by physicians and other healthcare providers trained in musculoskeletal medicine such as physiatrists, orthopedic surgeons, interventional radiologists, and sports medicine physicians, as well as rheumatologists. Commonly used agents include prednisolone, hydrocortisone, betamethasone acetate and betamethasone sodium phosphate, dexamethasone, methylprednisolone acetate, triamcinolone acetate, and triamcinolone hexacetonide [14].

### 3.2. Efficacy of Intra-Articular Steroid Injections

There is strong evidence in support of the short-term efficacy of IACs in reducing the symptoms of KOA. A 2004 meta-analysis of 10 randomized controlled trials (RCT) concluded that IACs provide symptom improvement for up to 2 weeks [15]. Similarly, a 2009 systematic review of six trials found that IACs resulted in a clinically and statistically significant reduction in KOA pain 1 week after injection, and another 2004 meta-analysis of five RCT found that the effects lasted up to 3 to 4 weeks [16,17]. More recently, a 2021 meta-analysis of 15 RCT found that IACs reduced pain and improved function early after administration for up to 6 weeks [18]. Despite this, there is no evidence of the long-term efficacy of IACs for KOA, nor is there consensus of the most efficacious agent [19,20]. Recent clinical trials of novel extended-release formulations of triamcinolone acetonide report statistically significant symptom reduction from 12 to 24 weeks [21,22]. More long-term studies are needed to fully characterize the true benefit over immediate acting IACs.

### 3.3. Complications of Intra-Articular Steroid Injections

While generally considered safe, IAC injections can occasionally lead to complications. Potential adverse effects include infection, post-injection flare, and changes in skin color at the injection site [23]. Concerns over accelerated OA progression and bone loss with long-term IACs injections have persisted since the advent of this therapy. A 2023 systematic review and meta-analysis of RCT and observational studies (*n* = 1437 patients) found that IACs had more cartilage loss on MRI than placebo [24]. Similarly, a 2020 case–control study found a reduction in meniscal thickness on MRI but no changes in the underlying cartilage or bone [25]. As previously discussed, the imaging findings may not necessarily correlate with clinical KOA symptoms, thus the clinical relevance of these findings is currently unknown.

### 3.4. Current Recommendations and Conclusion on Intra-Articular Steroid Injections

The Academy of the American Family Physician (AAFP) recommends IACs, but due to the potential risk of accelerated osteoarthritis the American Academy of Orthopedic Surgeons (AAOS) has downgraded IAC injections from strong to a moderate recommendation [10,26]. The Osteoarthritis Research Society International (OARSI) has a conditional recommendation for IACs to assist with the treatment of acute (1–2 weeks) and short-term (4–6 weeks) pain [27].

In conclusion, IAC injections provide short-term relief of pain and inflammation in patients with KOA. The recommended interval between IAC injections is three months [28] and patients scheduled to undergo total knee arthroplasty should not receive IACS within 3 months prior to the surgery [29]. While evidence supports their use for immediate symptom management, the long-term benefits and potential for complications necessitate a judicious approach to patient selection and treatment planning.

## 4. Intra-Articular Hyaluronic Acid Injection

### 4.1. Background and Mechanism of Action of Intra-Articular Hyaluronic Acid Injection

Hyaluronic acid (HA) is a non-sulfated glycosaminoglycan, composed of repeating disaccharide units of D-glucuronic acid and N-acetylglucosamine [30]. HA is found in various biologic tissues and serves an important role in the synovial fluid of articular joints such as the knee. The viscoelastic properties of HA enable it to act as a lubricating agent when the knee undergoes low shear stress activities and as a shock-absorbing elastic solid during high shear stress movements [31]. Furthermore, HA functions as a chondroprotective and anti-inflammatory agent [32]. In KOA, there are lower HA concentrations as it is depolymerized to a lower molecular weight and undergoes dysregulated synthesis, clearance, and degeneration within the affected joint [31,33]. Thus, the loss of sufficient amounts of HA in the synovial fluid in KOA results in the progressive degeneration of both articular and bony surfaces. The proposed mechanism of action of intra-articular hyaluronic acid (IAHA), also known as viscosupplementation, is to re-establish synovial fluid HA concentrations and stimulate endogenous production, which would reduce intra-articular friction, increase the synthesis of proteoglycans and aminoglycosides to support chondrocytes and cartilage matrix, and reduce the reactive inflammatory response associated with KOA [34].

### 4.2. Efficacy of Intra-Articular Hyaluronic Acid Injection

Clinically, IAHA has been utilized to treat mild to moderate OA when conservative management including IACS and physical therapy have failed, but data to support its use have been mixed. A 2011 meta-analysis [35] included 54 RCT with 7549 patients that compared IAHA to placebo in the treatment of KOA between 1983 and 2009, in which the primary outcome measure was a reduction in pain and secondary outcome measures were function and stiffness. The study found that IAHA became therapeutically effective at 4 weeks with a peak effect at 8 weeks and overall effect lasting until 24 weeks. Furthermore, the study concluded that the therapeutic benefits of IAHA exceeded the minimally clinically significant threshold and could be a viable potential therapeutic option for patients with mild to moderate KOA.

This conclusion was not supported by a more recent 2022 meta-analysis [36]. In this study, investigators included 169 RCT with 21,163 patients between 1972 and 2021 that compared IAHA to placebo to evaluate the efficacy of viscosupplementation to treat KOA. The study found that IAHA had a small, non-clinically relevant reduction in pain intensity and improvement in function compared to placebo. Additionally, IAHA was associated with a statistically significant risk of adverse events compared to placebo. Thus, the study concluded that while IAHA was found to reduce pain and improve function, the marginal benefits did not exceed the minimally clinically important difference (MCID) compared to placebo and did not support the broad use of IAHA to treat KOA. MCID is a patient-focused measurement used to determine the effectiveness of an intervention, as it is the smallest amount of improvement that would still be considered worthwhile by a patient [37].

### 4.3. Complications of Intra-Articular Hyaluronic Acid Injection

IAHAs are generally considered to be safe, but they are associated with an increased risk of local transient reactions when compared to placebos [38]. According to the American College of Rheumatology, local reactions typically involve redness, swelling, and temporary stiffness following the injection, but these symptoms typically resolve over time [39]. As with all interventions that involve breaching the skin, there is also a small risk of infection, but this is rare. Inflammation or infection within 3 days of IAHA was found to be <0.003% [40].

### 4.4. Recommendations and Conclusions on Intra-Articular Hyaluronic Acid Injection

The AAFP, which has based its recommendations on those of the American College of Rheumatology and Arthritis Foundation, and AAOS both do not recommend IAHA as a treatment for KOA. Interestingly, the OARSI has conditionally recommended IAHA for longer-term therapeutic pain relief due to its association with symptom amelioration beyond 12 weeks [27].

While there is evidence that IAHA reduces pain and stiffness as well as increases knee functionality, these improvements are not clinically substantial enough to be seen as a viable treatment for patients with KOA. The utilization of IAHA should be evaluated on a case-by-case basis with mutual decision-making between the patient and provider.

## 5. Intra-Articular Platelet-Rich Plasma Injection

### 5.1. Background and Mechanism of Action of Intra-Articular Platelet-Rich Plasma Injection

Platelet-rich plasma (PRP) is an autologous blood product used in regenerative medicine, in which the patient’s blood is taken, prepared through centrifugation, and plasma with a higher platelet concentration than seen at physiological levels is reintroduced into the patient. Platelets serve numerous functions within the body, but their role in facilitating healing has been explored as a treatment option for OA. While the mechanism of action is not fully understood, it has been proposed that growth factors produced by platelets such as platelet-derived growth factor (PDGF), vascular endothelial growth factor (VEGF), transforming growth factor-β (TGF-β), and epidermal growth factor (EGF) are involved in healing and anti-inflammatory cascades that create a microenvironment to counter the degeneration seen in OA [41,42]. PDGF has been shown to inhibit osteophyte formation, reduce inflammation, and promote cartilage repair [43]. Interestingly, while VEGF is involved in the vascularization of cartilage, it has also been associated with pathological processes of OA such as the development of osteophytes [44]. TGF-β is associated with tissue formation and repair as well as cell proliferation [45]. Lastly, EGF plays an important role in the maintenance of the superficial articular cartilage layer [46]. The overall therapeutic premise of PRP is to facilitate endogenous healing within the joint through supraphysiologic levels of growth factors produced by concentrated platelets.

There are several different formulations that are used for PRP injections, which vary depending on the commercial system that is used [47,48,49]. As a result, there is no consensus on the composition of the PRP that ends up being reintroduced into the patient as different commercial formulations have differing concentrations of supraphysiologic platelets. Platelet concentrations [50] range from 300,000 per mm^3^ to 1,500,000 per mm^3^. Further, individual patient characteristics such as medications are taken into account during the commercial preparation phase that impact the characteristics and composition of the PRP [51]. There are two general types of PRP, based on whether there are fewer leukocytes in the PRP compared to whole blood, which is known as leukocyte-poor PRP (LP-PRP), or whether there are more leukocytes, which is known as leukocyte-rich PRP (LR-PRP).

In a recent prospective double-blind RCT [52], patients were either treated with three intra-articular injections of LR-PRP or LP-PRP. The primary outcome was evaluated via changes in the International Knee Documentation Committee (IKDC) score 12 months after patients were treated. The IKDC scores assess patient symptoms, function, and sports activities on a scale of 0–100, in which 100 reflects the highest functioning and sports activity and the lowest degree of symptoms [53]. There were significant improvements in mean IKDC scores from the LR-PRP (45.6 ± 15.5 to 60.7 ± 21.1; *p* < 0.0005) and LP-PRP (46.8 ± 15.8 to 62.9 ± 19.9; *p* < 0.0005) groups at the 12-month time point, but there was no significant difference between the groups (*p* = 0.626). The study concluded that the LR-PRP and LP-PRP resulted in no clinical difference in outcomes.

### 5.2. Efficacy of Intra-Articular Platelet-Rich Plasma Injection

Similar to IAHA, PRP has been used for the treatment of mild to moderate KOA with a variable consensus on its efficacy. In a double-blind RCT [54], 78 patients with bilateral knee OA (156 affected knees) were randomized into 3 groups: (A) single LP-PRP injection; (B) two LP-PRP injections occurring 3 weeks apart; (C) control group with normal saline injection. Participants were asked to fill out a Western Ontario and McMaster University’s Arthritis Index (WOMAC), in which patients rated their pain, stiffness, and physical function at the start of the trial to establish a baseline and again during subsequent follow-ups (3 weeks, 3 months, and 6 months). There was a significant improvement (*p* < 0.001) for all subcategories of WOMAC (pain, stiffness, and physical function) as well as total WOMAC scores in both groups that received LP-PRP injections (groups A and B). There was significant worsening of pain, physical function, and total WOMAC scores in the control group C (*p* < 0.05). Interestingly, there was no significant difference in the degree of improvement between group A and B; thus, the study posited that a single LP-PRP injection is as efficacious as two injections in the treatment of early-stage KOA.

In a more recent study [55] through the RESTORE RCT, 288 patients with mild KOA received either a series of 3 LP-PRP or normal saline injections spaced one week apart. Patients were followed for 12 months and assessed on two primary outcomes: (1) average pain scores (11-point scale; 0–10 with higher numbers indicating worse pain) and (2) medial tibial cartilage volume evaluated through MRI. There was improvement in pain scores in both the LP-PRP and saline groups with the mean change [standard deviation] in pain scores for LP-PRP being −2.1 [SD 2.7] and saline being −1.8 [SD 2.5]. The MCID for an 11-point pain scale was defined as a needed change of at least 1.8. Thus, both groups met this therapeutic criterion. The mean difference between the groups favored LP-PRP but was not found to be significant (−0.4 points [95% CI, −0.9, 0.2]). Furthermore, the mean change in medial tibial cartilage volume for LP-PRP was −1.4% [SD 7.2%] and saline was −1.2% [SD 7.2%] with no statistically significant mean difference between the groups (−0.2% [95% CI, −1.9% 1.5%]). As a result, the findings of the study did not support the use of LP-PRP injections to treat mild to moderate knee OA.

### 5.3. Preparation of Intra-Articular Platelet-Rich Plasma Injection

There are two general methods to prepare PRP from the patient’s blood: (1) the PRP method, and (2) buffy coat method [56]. Generally, for the PRP method, the patient’s blood is drawn into tubes with anticoagulation factors and undergoes centrifugation through a “soft spin” to separate the whole blood into three layers from superior to inferior: (1) platelets and white blood cells (WBC), (2) buffy coat with majority WBC, and (3) red blood cells (RBC). For LP-PRP, the superior layer and superficial buffy coat layer are drawn off and placed into a new sterile tube. For LR-PRP, both the superior layer and entire buffy coat layer are drawn off. The tubes are then centrifuged again through a “hard spin” to allow for the formation of a platelet pellet at the bottom of the tube. The supernatant above the platelet pellet is drawn off and the remaining pellet is resuspended in a minimum quantity of plasma. For the buffy coat method, the whole blood is spun down at a “high speed” to create three layers from superior to inferior: (1) platelet-poor plasma, (2) platelets and WBC, and (3) RBC. The superior layer is drawn off and discarded. The intermediate layer is drawn off into a new sterile tube. For LR-PRP, no further steps are required. For LP-PRP, the contents of the tube can be passed through a leukocyte filter, or the tube can undergo low speed centrifugation to separate the platelets from the WBC.

### 5.4. Complications of Intra-Articular Platelet-Rich Plasma Injection

PRP injections are generally well tolerated with minimal to minor side-effects that are also commonly associated with other forms of intra-articular injections. Possible minor side-effects have been swelling, pain, and redness at the injection site, as well as transient, self-limited nausea, headache, and dizziness [50,57].

### 5.5. Recommendations and Conclusions on Intra-Articular Platelet-Rich Plasma Injection

The mixed results of several trials have resulted in the AAFP not recommending PRP injections as a therapeutic option for KOA. The AAOS formerly held the stance of strongly recommending PRP injections, but similarly, their guidelines have downgraded to limited recommendation for the use of PRP injections in the treatment of OA [10]. The OARSI does not recommend PRP injections, as the society has found the supportive evidence of this therapy to be of low quality [27,58].

In conclusion, there are many factors that need to be considered when treating KOA with PRP, including the type of PRP (LP vs. LR), concentration of platelets within the PRP, and number of injections. Furthermore, PRP is not generally covered by insurance companies in the United States, which may pose a financial burden to patients. Given the mixed consensus of the efficacy of PRP, there needs to be further research with more robust evidence to support its use.

## 6. Technique and Approach to Intra-Articular Injections

There are several different anatomic sites, approaches, and aids (blind vs. imaging-assisted) that are considered when administering intra-articular knee injections. Broadly, there are eight anatomic sites [59] through which the intra-articular space of the knee can be accessed: (1) lateral suprapatellar, (2) superolateral patellar, (3) lateral midpatellar, (4) anterolateral joint line, (5) infrapatellar, (6) superomedial patellar, (7) medial midpatellar, (8) anteromedial joint line. Regarding approaches, the knee can be flexed to various degrees, typically 30°, 45°, or 90°, or fully extended [59]. Guided intra-articular knee injections can be performed in conjunction with ultrasound, fluoroscopy, and air-arthrography.

In a 2013 systematic review of 23 studies [59], image-guidance improved the accuracy of intra-articular knee injections, but blinded injections at any of the lateral sites had enough reasonable accuracy to enable clinician preference to be practiced. For blind injections, superolateral patellar was the most accurate (87%), closely followed by lateral midpatellar (84%), lateral suprapatellar (83%), and anterolateral joint line (70%). The most common approach to knee injections is the anterolateral approach. First, a short-acting anesthetic is mixed with an appropriately dosed corticosteroid and loaded into a syringe. The patient is seated with the knee flexed at 90° and the patella, patellar tendon, tibial tuberosity, are identified and marked. The anterolateral soft spot is identified and marked. Topical analgesia is then applied to the area followed by sterile preparation of the skin. The needle should be guided from the anterolateral soft spot towards the intercondylar notch. Resistance should be felt at the skin then at the joint capsule. Prior to injecting the steroid/anesthetic mix, aspiration can be conducted to ensure no vascular structures are compromised [60]. Image guidance with ultrasound or fluoroscopy can be used to ensure accurate placement.

## 7. Genicular Artery Embolization

### 7.1. Background and Mechanism of Action of Genicular Artery Embolization

The genicular arteries (GA) originate from the superficial femoral artery and popliteal arteries. They are composed of six main branches: descending, superior medial, superior lateral, recurrent anterior tibial artery, inferior medial, and inferior lateral GA. The genicular arteries supply numerous structures in the knee including the synovial lining. The chronic inflammation associated with KOA results in pathologic neo-angiogenesis and the formation of demyelinated sensory nerve fibers, which contribute to structural changes and pain [61,62]. The mechanism of action behind genicular artery embolization (GAE) is to selectively embolize aberrant arteries supplying the hypervascularized regions of the knee that correlate with areas of pain to restrict OA-induced synovial inflammation [61,63]; however, a recent subgroup analysis of a RCT suggests that complete embolization of all GAs may improve outcomes [64]. Genicular artery embolization (GAE) was first introduced as a potential therapy for KOA in 2014 by Dr. Yuji Okuno. In a landmark study [65], 14 patients underwent GAE with significant improvement in mean WOMAC pain and overall WOMAC scores as well as the mean overall Visual Analogue Scale (VAS) pain scores. All 14 procedures were found to have achieved 100% technical success, which was defined as the reduction or cessation of blood flow in the intended vascular bed that was targeted.

### 7.2. Efficacy of Genicular Artery Embolization

GAE is a nonsurgical intervention used to treat KOA refractory to conservative therapies in patients, who are poor surgical candidates for TKA or those that prefer nonsurgical management but have failed other forms of conservative treatment. As GAE remains a relatively new intervention, there are limited data, but there is a consensus regarding the utility of GAE as a treatment for KOA. In a recent 2023 systematic review and meta-analysis [66], 9 studies including 337 knees evaluated the clinical outcomes of GAE related to retreatment rate, technical success, VAS, and WOMAC scores, with the latter two being evaluated through weighted mean difference (WMD) from baseline. The retreatment rate was evaluated at the 2-year follow up and compared to the rate of retreatment seen in total knee replacement (TKR) within the same time frame. The rates of retreatment were comparable with a rate of 8.3% (95% CI; 5.6–12.0%) for GAE and 5.2% (95% CI; 3.1–8.4%) for TKR. The technical success of GAE was defined differently by the included studies, but overall, it was determined to be 99.7% (95% CI; 97.1–100%). Treatment of KOA with GAE resulted in significant improvement in symptoms. At 12 months, there was a WMD for VAS and total WOMAC scores of −36 (95% CI: −51 to −22; *p* < 0.001 vs. baseline) and −34 (95% CI: −39 to −30; *p* < 0.001 vs. baseline), respectively. The study concluded that GAE is a potentially efficacious treatment for KOA given the limited evidence so far.

### 7.3. Genicular Artery Embolization Technique

GAE is performed by interventional radiologists as an elective outpatient procedure. The following outline of the procedure is summarized from a recent technical article [67]. The typical steps conducted prior to a routine angiogram should be followed (i.e., obtaining consent and basic laboratory tests, fasting before the procedure, and holding oral anticoagulant agents in case of need). Although not required, patients may be asked to identify the region(s) of the knee that cause the most pain and these sites are marked with a radiopaque marker. According to local protocol, the intervention can be conducted under moderate sedation, and the patient receives IH heparin to prevent clotting in small genicular arteries. The addition of antibiotics, NSAID, and steroids can vary according to the center, and there is no consensus on these adjunctive treatments.

The patient is positioned supine. Using a sterile technique, local anesthesia is administered before to obtain antegrade vascular access into the ipsilateral femoral artery, contralateral femoral artery, or radial artery, depending on operator preference. Ultrasound-guided puncture is commonly used. The control lateral approach or pedal approach can be proposed in selected cases. A 3 to 5 F catheter is inserted either bare back or through an introduced sheath. A digital subtraction angiogram (DSA) is conducted to identify the genicular arteries (Figure 3). Abnormal synovial vasculature can be identified by a blush, best visualized in the delayed phase of the DSA. Cone-beam CT can also be used to identify the genicular arteries. To access the genicular arteries, microcatheters that are 2 Fr or smaller are typically used. Collateral and/or cutaneous branches are avoided with the use of selective DSA to better visualize target branches. An injection of 100–200 mcg of nitroglycerin intra-arterially can be used if permanent particles are used, once the microcatheter is positioned at the desired location to improve blood flow to the pathologic hypervascularized synovium. In the same situation, cutaneous branches of the GA are vasoconstricted with an ice pack prior to the embolization of the target GA. Various embolic agents have been used, from rapidly resorbable ones (Imipenem or Lipiodol emulsion) to gelatin or other resorbable embolics [65,68,69,70]. Under fluoroscopic guidance, the embolization agent is slowly injected. The patient should recover in the postoperative area for 2–4 h with the procedure leg in full extension. A short course of an NSAID and steroid taper can be prescribed to minimize postoperative pain and inflammation according to local protocols.

### 7.4. Complications of Genicular Artery Embolization

GAE has been generally regarded as a safe procedure with minor associated complications most commonly pertaining to pain, self-resolving transient changes in skin color, and hematomas at the access site [71]. More major complications have been rarely reported including temporary paresthesia, bone infarctions, and fat necrosis [72].

### 7.5. Recommendations and Conclusions on Genicular Artery Embolization

Because GAE is such a new therapy to treat KOA, there are no specific recommendations from either AAFP, AAOS, or OARSI. Despite there being no formal recommendation from medical societies, clinical research studies have demonstrated that GAE is a viable non-surgical option for patients with KOA refractory to medical therapy, but more placebo/sham-controlled studies are needed to help validate GAE.

## 8. Genicular Nerve Ablation

### 8.1. Background and Mechanism of Action of Genicular Nerve Ablation

The joint capsule of the knee is innervated by the branches of several lower extremity nerves including the sciatic, obturator, and femoral nerves [73]. The articular branches of these nerves are called the genicular nerves (GN) [74]. While there is some debate about the exact anatomy of the GN, they are generally composed of six branches: superior lateral, superior medial, inferior lateral, inferior medial, suprapatellar, and recurrent tibial GN. Radiofrequency neurotomy is a minimally invasive technique used to treat chronic pain, which utilizes the energy from radio waves to ablate nerves preventing pain signal transmission [75]. Dr. Woo-Jong Choi is attributed to pioneering the use of radiofrequency neurotomy to the genicular nerves as a potential therapy for KOA in 2011. In a double-blind RCT [74], investigators preformed radiofrequency neurotomies on the GN of 38 patients, who had a positive response with a diagnostic GN block and ≥3 months of severe OA-associated pain that was not relieved with conservative management. There was significant improvement in VAS pain scores (*p* < 0.001) and Oxford knee scores (*p* < 0.001). While Dr. Choi first demonstrated the potential efficacy of GN neurotomies with radiofrequency waves, there are additional methods that can be utilized in nerve ablation such as cryoablation [76] and chemical ablation [77].

The mechanism of action of genicular nerve ablation (GNA) is to halt the transmission of pain signals caused by KOA through the selective destruction of causative genicular nerves (Figure 4).

### 8.2. Efficacy of Genicular Nerve Ablation

Similar to GAE, GNA is a nonsurgical minimally invasive treatment for patients with moderate to severe KOA, who are poor surgical candidates or have failed conservative management and/or surgery [78]. Clinical trials have been generally supportive of the efficacy of GNA in comparison to other forms of treatment for KOA [79]. In a single-blind RCT, 60 patients with chronic KOA (radiographic KL classification of stage 3 or 4) were randomized to receive either conventional analgesics alone or radiofrequency neurotomy of the GN. The primary outcomes of this study were VAS for pain and WOMAC for disability. The baseline scores were established at the initial visit. Interval check-in assessments were performed after the prescribed intervention at the 2-week, 3-month and 6-month time points. The patients that were assigned to the conventional analgesics group were given oral NSAIDs and paracetamol with physical therapy if needed. Patients who underwent the GNA experienced significant improvement compared to the control for VAS scores at the 2-week, 3- and 6-month check-in time points (*p* < 0.005) as well as at the 6-month time point for WOMAC scores (*p* < 0.001). The study concluded that GNA was an effective treatment for patients with moderate to severe KOA that is refractory to conservative treatment.

### 8.3. Genicular Nerve Ablation Technique

GNA is an elective outpatient procedure that can be conducted under local skin anesthesia in an imaging suite or conscious sedation in an operating room. It can be performed by numerous different specialists such as physiatrists, interventional radiologists, anesthesiologists, neurologists, and some subspecialities of surgery. The following technique pertains to the application of radiofrequency ablation of the GN and is a summary of the procedure outlined by a 2019 technical article [78]. Before undergoing GNA, a diagnostic GN block with local anesthetic is performed. Patients who have a positive response are appropriate candidates for GNA. A positive response is defined as having ≥50% relief of pain.

The patient is positioned supine with the procedural knee flexed to 30°. A grounding pad is placed on the contralateral lower extremity or ipsilateral upper or lower extremity. Under sterile technique with fluoroscopy or ultrasound, the entry points are visualized and cutaneously marked at the base of the lateral and medial condyles of the femur (i.e., approximated anatomic locations for the superior lateral and superior medial GN, respectively) and base of the medial condyle of the tibia (i.e., approximated anatomic location of the inferior medial GN). A total of 1–2% Lidocaine is used to create a wheal at the cutaneous region of the three planned sites. Under anteroposterior fluoroscopy, three cannulas (20-gauge, 3.5 in) with 10 mm active tips are positioned at the marked sites that correlate to the location of the superior lateral, superior medial, and inferior medial GN. The cannulas at each site should be advanced until the tip is in contact with bone. Obtain a lateral fluoroscopic image to ensure proper cannula placement, which should be at the diaphysis midpoint of the tibia and femur. Reposition as necessary to achieve proper positioning. Prior to ablation, sensory confirmation can be obtained by stimulating the region at ~50 Hz until knee pain is replicated. Confirm there is no motor stimulation with administration of 2 Hz and 2 V at each site. Reposition and retest as needed until there is no motor activation. Withdraw the cannulas 2–3 mm, inject 1 mL of 1% Lidocaine, and wait 90 s for the anesthetic effect. After waiting the appropriate amount of time, the radiofrequency electrodes are inserted into the cannulas. The temperature of each electrode should be turned to 80 °C for 1 min. Each electrode should be advanced 3–5 mm at each site and a second round of ablation at the same settings should be performed. Postoperative pain can be mitigated by administering 1 mL of methylprednisolone into each site via the respective cannula. Cannulas are then removed, and bandages are applied to the sites. A fourth cannula can be used to place a probe for ablation of the suprapatellar genicular nerve [80].

### 8.4. Complications of Genicular Nerve Ablation

GNA is considered a safe procedure with minor symptoms such as pain and tenderness at the access sites, but it has been associated with rare major vascular adverse events and complications including osteonecrosis of the patella, formation of arteriovenous fistula, hemarthrosis, and pseudoaneurysm [81].

### 8.5. Recommendations and Conclusions on Genicular Nerve Ablation

The AAFP and OARSI have no specific recommendations for GNA. Regarding recommendation status from the AAOS, there is limited recommendation for “denervation therapy”, which encompasses a much larger therapeutic range than GNA. AAOS recommendations were “downgraded two levels due to inconsistent evidence and bias” associated with a few studies that did not explicitly involve GNA [10,82], so it is challenging how to interpret this recommendation in regard to GNA. Despite this, clinical studies have supported the efficacy of GNA as being a viable non-surgical option for the treatment of moderate to severe KOA.

## 9. Genicular Nerve Cryoneurolysis

### 9.1. Background and Mechanism of Action of Genicular Nerve Cryoneurolysis

Cryoneurolysis is the use of cryoablation probes to ablate nerves in accordance with the Sunderland classification [83,84]. Current best practice is to achieve a Sunderland category 2 neurolysis (myelinolysis and axonolysis) which typically requires temperatures between −20 and −100 degrees Celsius [85,86,87], although the duration of exposure is also important in addition to absolute temperatures. Clinically available cryoablation systems in the United States are not able to reliably achieve lower than −100 degrees Centigrade and therefore the current clinical best practice is to ensure ablation in excess of a Sunderland 1 (which can amplify pain rather than reduce it).

There are benefits and disadvantages to cryoneurolysis compared to radiofrequency neurolysis and chemical neurolysis. The main benefit is that cryoneurolysis is generally less painful during and after the procedure and is appropriate for those patients who cannot tolerate other modalities. Another advantage is the larger zone of ablation achieved with cryoneurolysis, which can cover the diverse locations of the geniculate nerves in the antero-posterior axis at the bases of the femoral/tibial condyle, respectively. Cryoablation can be helpful in patients with significant metal instrumentation or electronic implants that may deter monopolar RFA despite a grounding pad (for example, pacemakers, spinal cord stimulators). The visualization of the ice on CT imaging or ultrasound confirms that the target area has been adequately covered.

The main disadvantage is cost, as cryoprobes are significantly more expensive than radiofrequency probes. Additionally, most cryoablation units require capital equipment purchase as well as purchase of argon gas. This can be overcome by using a smaller hand-held cryoablation unit, which is commercially available but remains a capital equipment expense. Currently commercially available cryoablation units do not have an on-board nerve stimulator, which is a relative disadvantage as motor stimulation cannot be ruled out prior to ablation. This can be overcome by using a relatively inexpensive independent nerve stimulator, though this is an additional capital equipment cost. Another practical solution is the performance of a diagnostic nerve block as a separate session procedure with confirmation of a lack of motor functional degradation immediately post-block (before the block diffuses) and then adhering to those landmarks for the cryoneurolysis. Finally, because an intact cortex is only resistant to impedance-based ablation modalities like radiofrequency ablation [88], cryoablation can create small zones of bone ablation adjacent to the nerve, which remain understudied and are of unknown but doubtful clinical significance. Finally, while the larger zone of ice can be advantageous for the condylar geniculate nerves, it is relatively overpowered for the patellar nerve which runs the risk of skin injury with currently available cryoablation probes.

### 9.2. Efficacy of Genicular Nerve Cryoneurolysis

Geniculate cryoneurolysis remains understudied. In a randomized controlled trial, Mihalko and colleagues reported reduced pain with opioid use when using genicular cryoneurolysis as a pre-operative adjunct to total knee arthroplasty [89]. Nygaard and colleagues are studying management of chronic pain in patients with knee osteoarthritis with a double-blinded randomized controlled sham trial [90].

### 9.3. Genicular Nerve Cryoneurolysis Technique

The technique for cryoneurolysis is identical to radiofrequency neurolysis with the exception that only the superior medial, inferior medial, and superior lateral geniculate nerves are targeted. With identical probe positioning, a common freeze–thaw protocol is 10 min freeze, 3 min passive thaw, 3 min freeze, and 3 min passive thaw. Only passive thaws are performed in cryoneurolysis to avoid unnecessary and disorganized osmotic injury.

### 9.4. Complications of Genicular Nerve Cryoneurolysis

The complication profile of cryoneurolysis is similar to radiofrequency neurolysis but remains underreported. A unique complication of cryoneurolysis occurs with the under ablation of the nerve, which can generate a Sunderland 1 neurolysis and worsened pain, particularly with new onset of neuropathic pain in addition to the chronic nociceptive arthritis pain. This can be treated with gabapentin, pregabalin, oral corticosteroids, nerve block, and/or repeat cryoneurolysis.

### 9.5. Recommendations and Conclusions on Genicular Nerve Cryoneurolysis

The AAFP and OARSI do not have any specific recommendations for genicular nerve cryoneurolysis. As for the recommendation from the AAOS, genicular nerve cryoneurolysis is categorized under the broader umbrella of “denervation therapy”, which genicular nerve ablation also falls under. Thus, for the same reasons previously mentioned in the last section, there is limited recommendation for cryoneurolysis [10].

For centers with a cryoablation machine on-site, cryoneurolysis is a useful adjunct for patients who are unable to tolerate radiofrequency neurolysis, or those who prefer to avoid it due to the theoretical risk of monopolar radiofrequency energy interacting with pacemakers and/or spinal cord stimulators.

## 10. Regenerative and Experimental Therapies

Regenerative therapies have gained increasing attention from patients, clinicians, and researchers in the treatment of KOA due to the progressive nature of this disease. These therapies aim to reverse the degenerative process through the restoration of the extracellular matrix, the introduction or recruitment of mesenchymal stem cells, and/or the stimulation of dormant chondrocytes. The use of these therapies remains largely limited to experimental pre-clinical and early clinical studies due to concerns regarding their efficacy, safety, and ethical implications. Nevertheless, regenerative therapies represent a promising advancement in the management of KOA.

### 10.1. Intra-Articular Collagen Injections

Adult articular cartilage predominantly consists of collagen. Cross-linked collagen fibers form a three-dimensional network in the extracellular space near the surface of healthy cartilage [91]. In KOA, this collagen scaffold undergoes irreversible degradation. Intra-articular injections of purified bovine or porcine collagens aim to repair this damage [92]. A 2023 narrative review [93] concluded these injections may effectively stimulate chondrocytes to produce hyaline cartilage. The seven studies included in the review all reported short-term improvements in pain and function. In an RCT [94] comparing type-1 hydrolyzed collagen to HA, collagen demonstrated comparable efficacy and safety. The authors concluded that the lower cost of collagen makes it a viable alternative intra-articular therapy. However, further studies are needed to assess collagen’s capacity to recruit chondrocytes, particularly in severe OA where patients may have limited cellular reserves.

### 10.2. Mesenchymal Stem Cells

In recent years, mesenchymal stem cells (MSCs) have gained prominence in therapeutic applications for degenerative diseases [95]. The appeal of these pluripotent cells lies in their ability to differentiate into chondrocytes, regenerate spontaneously, and modulate the immune response to slow or even reverse the degenerative process [96,97]. MSCs are typically harvested from either bone marrow or adipose tissue, undergo clonal expansion ex vivo, and are subsequently introduced to the knee either through an intra-articular injection or percutaneous implantation [96].

A large 2018 meta-analysis [98] of 35 studies including 2385 patients evaluated the clinical outcomes of MSC therapies for KOA. Standardized mean difference (SMD) was used to assess the treatment effect on pain, physical function, and cartilage volume. The pooled SMDs on VAS knee pain exceeded the effect of IACs and NSAIDs. The mean differences for both pain and self-reported physical function were ≥10%, exceeding the MCID. However, the effect sizes were attenuated on sensitivity analysis using only RCTs. In a sub-analysis, autologous MSCs showed greater pain relief effects than allogeneic MSCs, while factors such as implantation (compared to injection) were associated with higher WOMAC SMDs, with rehabilitation emerging as a significant effect modifier, underscoring its influence on treatment outcomes. Similarly, a systematic review [99] reported significant improvements in VAS and WOMAC scores at 6 and 12 months post intra-articular MSC injections.

Bone marrow-derived mesenchymal stem cells (BM-MSCs) have been popular due to the high cell yield and proliferative activity [100]. However, concerns exist regarding the quality of cells harvested from patients with end-stage OA [96]. Alternatively, adipose tissue-derived MSCs (AD-MSCs) and stromal vascular fraction (SVF) are gaining popularity due to their resistance to hypoxia, greater anti-inflammatory properties, and maintenance of chondrogenic potential [100]. A recent 2023 meta-analysis [101] of 21 studies compared the efficacy and safety of AD-MSCs and BM-MSCs. The results indicate that both treatments were effective in reducing VAS scores at 6 months; however, the improvement persisted for up to 1 year with AD-MSCs but not with BM-MSCs. Additionally, AD-MSCs had a greater effect on WOMAC scores than BM-MSCs and fewer adverse events.

The efficacy of AD-MSCs and SVFs in KOA management is further supported by a 2021 meta-analysis of 18 studies [100] that reported a statistically significant improvement in WOMAC pain scores with progressive clinical improvement observed over follow-up periods of up to 18 months. Additionally, AD-MSCs are easily harvested and do not experience age-related cellular derangements, making them suitable for patients with advanced OA. Despite positive results, the clinical use of MSCs remains controversial due to the use of highly heterogenous studies with varying protocols and designs.

### 10.3. Hydrogels

Hydrogels are three-dimensional, cross-linked polymers with complex chemical and physical properties that can be altered to induce cartilage repair or augment current therapies. The use of hydrogels as a treatment for KOA is based on two main ideas: (1) the targeted and controlled delivery of exogenous substances (i.e., drugs, stem cells) and (2) the formation of a stable microenvironment suitable for cartilage repair [102]. An ideal hydrogel should be a biocompatible, long-lasting polymer that localizes and adheres to damaged cartilage [103]. It should stimulate repair while maintaining structural integrity under mechanical stresses before being absorbed by the body [104]. Hydrogels that are produced from synthetic sources like polyethylene glycol (PEG), polyvinyl alcohol (PVA), poly (2-methacryloyloxyethyl phosphorylcholine) (PMPC) have the benefit of being mechanically strong [105]. Conversely, biological sources such as polysaccharides like chondroitin sulfate (CS) and hyaluronic acid (HA) are components of natural cartilage and better facilitate its endogenous formation while proteins like collagen are highly abundant and play a crucial role in the structure of the extracellular matrix [105]. A notable application is the use of hydrogels as a vehicle for stem cells therapies in KOA. Hydrogels concentrate low-dose MSCs at the treatment site, enhancing cell survival time and retention while maintaining the therapeutic effect [106]. Additionally, stem cells suspended in hydrogels demonstrate enhanced immunomodulatory function [107].

In a 2017 phase I/II clinical trial [108], seven patients with KL grade 3 KOA underwent treatment with a hyaluronate hydrogel containing MSCs (HA-MSC) derived from human umbilical cord blood. The cartilage defects were examined arthroscopically, followed by exposure of the defect site through a small longitudinal arthrotomy. Multiple drill holes were made at the defects on the femoral condyle in which HA-MSCs were implanted. Investigators observed maturing cartilage repair arthroscopically at 12 weeks, persistence of regenerated cartilage on MRI at 3 years, and a clinically significant improvement that remained stable over 7 years. Similar findings have been reported in a recent 2023 case series [109] and a 2024 retrospective review [110] using the same technique, while a 2024 phase I dose escalation trial [111] concluded that the intra-articular injection of low- and medium-dose HA-MSCs was safe and effective at improving WOMAC (*p* < 0.001) and VAS (*p* < 0.001) scores at 6 months. Interestingly, one 2023 pre-clinical trial [112] found that an injectable collagen-based piezoelectric hydrogel promoted stem cell migration and chondrogenesis in vivo, while also stimulating cartilage repair in rabbit models of OA. Exciting advancements in cell therapy and tissue engineering alongside their integration with innovative biomaterials highlight the potential for new minimally invasive treatment options for KOA.

## 11. Role of Artificial Intelligence in Decision-Making and Patient Selection

AI has shown promise in revolutionizing treatment decision-making and patient selection in the realm of KOA. Some recent studies highlight the potential of AI in the automatic grading of knee radiographs and predicting arthroplasty outcomes [113,114]. The authors note that current AI algorithms face limitations such as the lack of external validation, inherent biases in clinical data, and the necessity for extensive datasets. Moreover, there is a paucity of papers incorporating clinical and demographic factors, although limited evidence suggests associations between baseline knee pain, Heberden nodes, varus alignment, certain serum markers, and KOA progression [115]. Despite these challenges, AI shows strong potential in the management of KOA. Incorporating demographic, clinical, and imaging variables into a well-designed machine learning model could likely offer valuable support for clinical decisions. Nevertheless, current limitations must be addressed before this tool can be integrated into practice.

Traditional diagnostic and management approaches for KOA often rely on clinical assessments and imaging modalities, yet their effectiveness in early detection and personalized treatment remains limited. In recent years, AI has emerged as a promising paradigm in healthcare, offering novel solutions for disease diagnosis, prognosis, and therapeutic interventions. Leveraging advanced machine learning (ML) algorithms, AI applications in KOA hold the potential to revolutionize disease management by providing more accurate and timely assessments, enabling personalized treatment strategies, and improving patient outcomes. This section explores the current landscape of AI in KOA, highlighting the key methodologies, challenges, and future directions in harnessing AI-driven innovations to address the unmet clinical needs in KOA diagnosis and management. Figure 5 shows AI-integrated methodologies for KOA.

As previously mentioned, MRI assessment of KOA in symptomatic patients involved correlating MRI findings with pain levels using plain film radiographs and Kellgren–Lawrence (KL) scores. In one study, various pain levels were reported, with joint effusion significantly linked to pain severity. MRI findings were statistically analyzed using the Fisher exact test to obtain *p*-values. MRI revealed abnormalities in cartilage, meniscus, and ligaments, alongside bone marrow lesions, osteophytes, synovitis, and effusions, offering a comprehensive evaluation of joint structure in KOA [116]. Table 3 summarizes the methods and their AI methods for KOA.

MRI serves as a tool for KOA assessment, capturing the structural changes and biochemical alterations in tissues. Despite conventional radiography’s limitations, it remains the primary modality in OA clinical trials, with emerging hybrid techniques like PET/MRI offering potential enhancements. In addition, AI, particularly deep learning with convolutional neural networks, shows promise in deepening our understanding of OA progression, augmenting conventional MRI assessment methods. Hybrid imaging modalities such as PET/MRI have emerged as new image analysis techniques [125].

Current research using ML explores the use of AI in the morphological [114] characterization, diagnosis, and treatment of OA, particularly in image analysis. These techniques lead to more effective and personalized treatment options, where AI aids on OA phenotypic analysis involving multimodal imaging from Multicenter Osteoarthritis (MOST) study [131]. A recent 2024 systematic review and meta-analysis revealed that AI algorithms outperformed clinicians in sensitivity (88% vs. 80%) and specificity (81% vs. 79%) on internal validation sets, with even higher reliability in external validation (sensitivity: 94%, specificity: 91%) when detecting OA, focusing on temporomandibular joint OA. The authors developed an automated diagnostic tool from cone-beam computed tomography (CBCT) images, showcasing high accuracy in detection [132]. Similarly, ML methods for KOA diagnosis highlight the potential for automated solutions across various phases of OA management [133]. AI models targeting OA aim to improve diagnosis efficiency, assess risk pre-symptom onset, and detect biomechanical changes for personalized interventions [114].

Radiographic imaging from X-rays, MRI, and CT scans aid experts in the diagnosis of KOA. Meta-analysis and meta-regression based on multiple parameters on pooled data were used for diagnostic metrics and to find potential heterogeneous sources, which showed that AI has a promising role as an adjunct to radiologists [113]. AI enhances personalized medicine approaches, e.g., multi-layer perceptron (MLP), utilizing diverse patient parameters and imaging data [117,134]. The utilization of AI in these areas may enable automated diagnosis by highlighting abnormal regions in a framework called OsteoGA [135]. This framework enables the detection and classification of severity levels based on the KL score estimation system using two models: (1) one model would distinguish normal (KL 0–I) from KOA (KL II–IV), and (2) another model would classify the severity as normal (KL 0–I), non-severe (KL II), or severe (KL III–IV) [118]. The KL grading system can be combined with measurements from MRIs of medial meniscus extrusion (MME) and cartilage thickness [128], or with digital X-ray images of joints using multiple deep neural network models [119]. These methods show promise in predicting early disease and severity classification. Some current research in AI models use Faster RCNN, ResNet-50 with transfer learning and AlexNet on digital X-ray images of knee joints [119], a U-net model and VGG11 encoder to extract joints from radiographic images [129]. ML models also showed promise in grading knee radiographs and predicting the need for surgery as well as forecasting postoperative outcomes [114]. Many deep learning methods, CNN, have been developed to enhance clinical workflow efficacy [136]. Still, challenges remain regarding the interpretability of AI models for image analysis [114,137].

AI plays a crucial role in conducting epidemiological studies and facilitating population screening for OA. Comprehensive reviews of ML applications in OA clinical care, particularly focused on KOA, underscore the potential of deep learning techniques in imaging analysis [137]. AI-based systems enhance agreement rates of KL grading, joint space narrowing, and sclerosis and osteophyte OARSI grade and improved the accuracy in KOA diagnosis, in turn, potentially improving standard care [120]. Similarly, AI algorithms applied to radiographs reveal the high prevalence rates of KOA, identifying key risk factors using KL and OARSI [121]. In addition, a review evaluating the diagnostic efficiency of AI models in detecting temporomandibular joint osteoarthritis (TMJOA) through radiographic image highlights AI’s potential in automating TMJOA diagnosis, further contributing to epidemiological studies and population screening efforts [122]. These findings emphasize AI’s significant contributions to enhancing epidemiological research through analysis of large datasets and population-level screening strategies for OA.

AI-driven predictions are instrumental in identifying therapeutic targets and developing treatments for OA. By targeting senescent chondrocytes, AI identifies novel targets, offering potential avenues for OA treatment involving Senescence-Associated Secretory Phenotype (SASP) for physiological functions lost due to inflammation and extracellular matrix degradation [138]. Semi-quantitative MRI, coupled with correlation analysis for MRI findings and varying degrees of reported pain, enhances structural tissue pathology and joint characterization in OA research, paving the way for improved assessment accuracy and efficiency [116,130].

Unaided and aided methods for determining the severity of OA are used to assess AI model reliability in comparison to clinical severity. A study provided a comparison (via Fisher Z-transformed correlations) for AI-unaided and AI-aided readings for a total of 10 independent readings (5 AI-unaided and 5 AI-aided) [123]. Transfer learning deep learning models and convolutional neural networks (CNN) demonstrate high accuracy in detecting KOA severity from radiographs, indicating the potential for automated classification tools [124,139]. In one analysis, eight pretrained deep neural network models were used to predict OA using KL scoring, ResNet50, VGG-16, InceptionV3, MobilnetV2, EfficientnetB7, DenseNet201, Xception and NasNetMobile [124]. ResNet architecture with a similar procedure used through another study led to an area under curve (AUC) of 0.65–1.00 corresponding to Osteonecrosis, Lateral, medial, and Patella X-ray images [139].

Applying AI in the management of KOA presents several challenges. A major limitation is the quality and quantity of data available for training AI models. These systems rely on large, high-quality datasets to make accurate predictions and recommendations. However, KOA datasets are often constrained by small sample sizes, variability in imaging techniques, and inconsistencies in diagnostic criteria, which can affect the generalizability and accuracy of AI models, leading to less reliable predictions and interventions. The complexity of KOA, with its diverse patient demographics and varying disease stages, further complicates the development of universal AI solutions. Additionally, the interpretability of AI findings is a significant hurdle. AI often produces black-box predictions without clear explanations of the reasoning behind them, which can undermine clinicians’ trust and their ability to make informed decisions based on these recommendations. Moreover, integrating AI tools into existing healthcare systems poses challenges related to interoperability, data privacy, and the need for clinician training. These barriers can hinder the effective implementation and utilization of AI in managing KOA, potentially limiting its benefits in improving patient outcomes.

### AI Recommendations

AI, particularly deep learning models like CNNs, has shown significant promise in enhancing the diagnosis and management of KOA. The traditional manual methods for diagnosing and annotating KOA are time-consuming and subject to variability, whereas AI approaches offer a more consistent and efficient alternative. The adoption of both 2D and 3D CNNs in the field has led to notable improvements in the accuracy and efficiency of KOA detection and progression/treatment monitoring. Specifically, 3D CNNs leverage the volumetric nature of medical imaging modalities such as MRI, providing a more comprehensive analysis of joint structures and changes over time. The reviewed studies highlight the potential of these technologies to transform clinical workflows, making them more streamlined and less dependent on manual intervention.

To maximize the benefits of AI in KOA, it is crucial to integrate these technologies seamlessly into existing clinical workflows. This includes training healthcare professionals to use AI tools effectively and ensuring that these tools complement, rather than replace, their expertise. Also, there is a need for standardized protocols and large-scale validation studies to ensure the reliability and generalizability of AI models across diverse populations and imaging equipment. Collaborative efforts among researchers, clinicians, and industry stakeholders are essential to establish these standards. For such analysis, encouraging data sharing and collaboration across institutions can enhance the robustness of AI models. Large, diverse datasets improve model training and validation, leading to more accurate and generalizable results. Moreover, AI models should be designed with patient-centric outcomes in mind, focusing not only on diagnostic accuracy but also on improving prognostic capabilities and treatment planning. This approach ensures that AI contributes meaningfully to patient care and quality of life. Continuous research and innovation are vital to keep pace with advancements in AI and medical imaging. Investing in interdisciplinary research that combines expertise in AI, radiology, and orthopedics will drive further breakthroughs in the field. By addressing these recommendations, the implementation of AI in KOA can be optimized, ultimately leading to improved patient outcomes, enhanced clinical workflows, and more personalized and effective treatments.

## 12. Conclusions

In conclusion, there are several promising advancements in minimally invasive treatment options for KOA, including intra-articular injections, genicular artery embolization, and genicular nerve ablation. While traditional treatment methods for KOA offer short-term relief, new methods have shown potential for more effective and longer-lasting outcomes. Furthermore, the integration of AI into the diagnostic and treatment stratification process further enhances the personalization of care. Further research efforts and clinical trials are essential to fully understand the efficacy and safety of these innovative approaches with the ultimate goal of improving patient outcomes and quality of life.

## Figures and Tables

**Figure 1 jpm-14-00970-f001:**
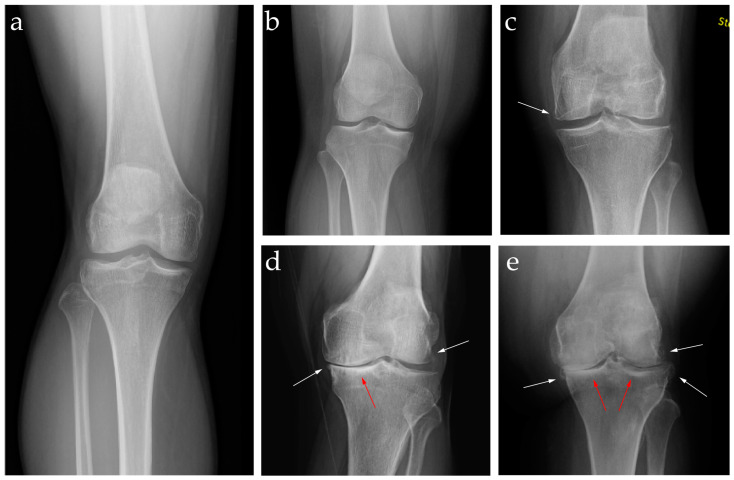
Weight-bearing anterior-posterior radiographs of knees in various stages of osteoarthritis. (**a**) KL grade 0—no radiographic features of OA; (**b**) KL grade 1—possible osteophytic lipping of the medial and lateral compartments and doubtful joint space narrowing; (**c**) KL grade 2—medial osteophyte (white arrow) with possible lateral joint space narrowing; (**d**) KL grade 3—multiple osteophytes (white arrows), definite narrowing of the medial compartment, and sclerosis of the medial tibial plateau (red arrow); (**e**) KL grade 4—multiple large osteophytes (white arrows), severe narrowing of the medial joint space, significant subchondral sclerosis (red arrows) although bone deformity is not present in this case.

**Figure 2 jpm-14-00970-f002:**
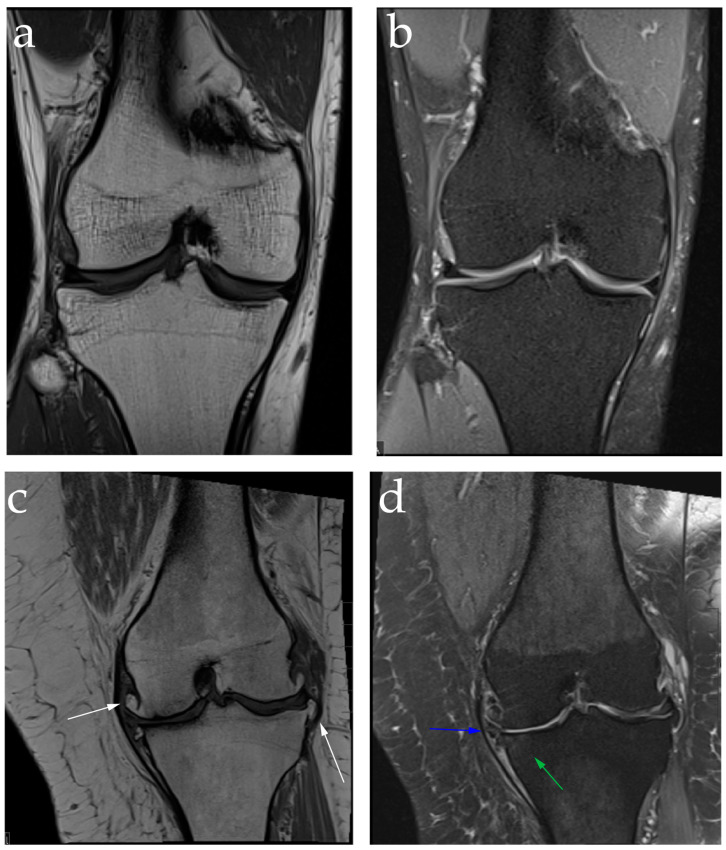
Coronal MR images of healthy (**a**,**b**) and osteoarthritic (**c**,**d**) knees. (**a**) T1-weighted and (**b**) fat-suppressed proton density-weighted images demonstrating no evidence of OA and normal soft tissue structures; (**c**) T1-weighted MR image showing joint space narrowing and multiple osteophytes (white arrows); (**d**) fat-suppressed proton density-weighted MR image demonstrating significant cartilage loss of the medial compartment and mild subchondral bone marrow edema of the medial plateau (green arrow) and extrusion of the body segment of the medial meniscus (blue arrow).

**Figure 3 jpm-14-00970-f003:**
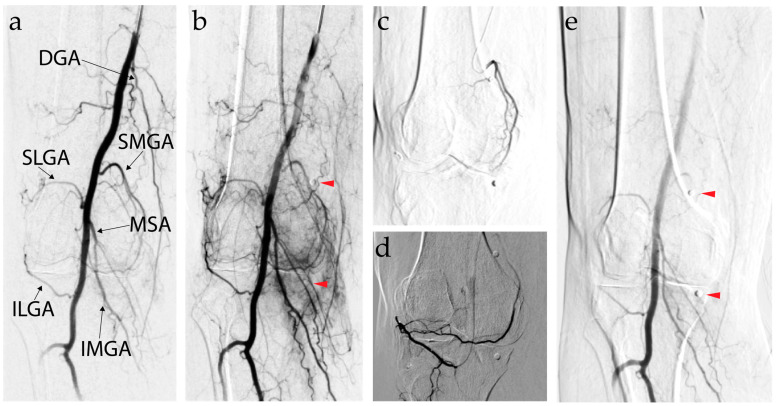
Annotated digital subtraction angiogram (DSA) depicting the genicular artery embolization with red arrowheads identifying radiopaque marker for the painful area. (**a**) Proximal popliteal artery early-phase angiogram showing the descending genicular artery (DGA), superior lateral genicular artery (SLGA), superior medial genicular artery (SMGA), medial sural artery (MSA), inferior lateral genicular artery (ILGA), and inferior medial genicular artery (IMGA); (**b**) delayed phase of DSA with inflammatory blush surrounding the painful areas identified with radiopaque markers (red arrowheads); (**c**) selective post-embolization DSA of SMGA; (**d**) selective post-embolization DSA of ILGA, since it was back supplying the SMGA angiozone; and (**e**) final post-embolization DSA from popliteal artery showing resolution of the inflammatory blush.

**Figure 4 jpm-14-00970-f004:**
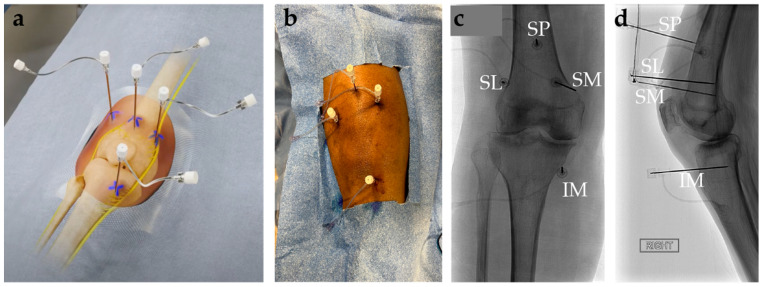
Targets and approach to genicular nerve ablation (GNA) procedure. (**a**) Schematic of knee innervation with target areas distinguished by X; (**b**) picture of the sterile field during GNA procedure; (**c**) fluoroscopic AP; and (**d**) lateral view of left knee demonstrating needle placement during the four-probe approach to GNA. Probes are placed near the bone to target the suprapatellar (SP), superomedial (SM), superolateral (SL), and inferomedial (IM) nerves.

**Figure 5 jpm-14-00970-f005:**
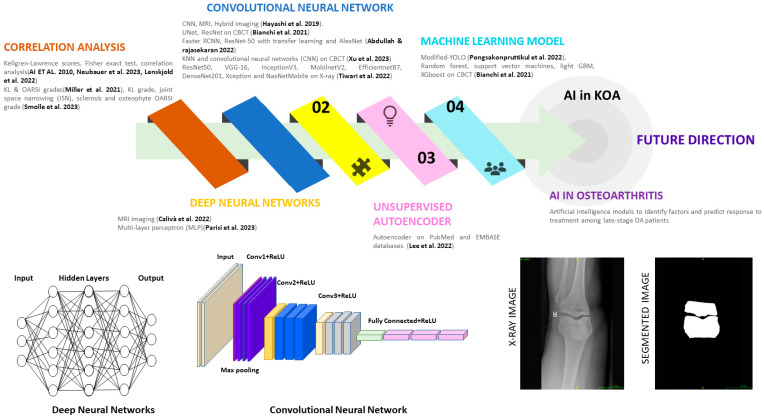
Artificial intelligence methodology for knee osteoarthritis [38,114,116,117,118,119,120,121,122,123,124,125,126,127].

**Table 1 jpm-14-00970-t001:** Kellgren and Lawrence grading system for radiographic osteoarthritis.

KL Grade	Grade 0	Grade 1	Grade 2	Grade 3	Grade 4
Classification	Normal	Doubtful	Mild	Moderate	Severe
**Description**	No radiographic features of OA	Minimal joint space narrowing with possible osteophytes	Evidence of at least one osteophyte with definite joint space narrowing	Multiple osteophytes, definite joint space narrowing and some evidence of sclerosis with or without deformity of the bone contour	Large osteophytes, complete obliteration of the joint space, severe sclerosis, and definite deformity of the bone contour

**Table 2 jpm-14-00970-t002:** Summary of management options for knee osteoarthritis.

Invasiveness	Type	Method
**Non-invasive**	Nonpharmacologic	Weight loss if overweight or obese (5–10% of body weight)
		Physical therapy
	Pharmacologic	oral analgesics (e.g., oral NSAIDs, duloxetine)
		topical analgesics (e.g., NSAIDs, capsaicin)
**Invasive**	Operative	arthroscopic procedures
		total knee arthroplasty
	Minimally invasive	Corticosteroid injections
		Hyaluronic acid injections
		Platelet-rich plasma injections
		Genicular nerve ablation
		Genicular artery embolization

**Table 3 jpm-14-00970-t003:** Summary of artificial intelligence methods for knee osteoarthritis.

References	HD Data Analyzed	Methods	Multivariate Data
AI et al., 2010 [116]	MRI	Kellgren–Lawrence scores, Fisher exact test, correlation analysis	-
Hayashi et al., 2019 [125]	MRI	CNN	-
Mohammadi et al., 2023 [113]	X-ray	Meta-regression	-
Bianchi et al., 2021 [117]	CBCT	Random forest, support vector machines, light GBM, XGboost, UNet, ResNet,	-
Parisi et al., 2023 [127]	-	Multi-layer perceptron (MLP)	The UARTA star-rating quality assessment scale
Pongsakonpruttikul et al., 2022 [118]	X-ray	Modified-YOLO	-
Sekiya et al., 2023 [128]	MRI	Statistical analysis	-
Abdullah and Rajasekaran 2022 [119]	X-ray	Faster RCNN, ResNet-50 with transfer learning and AlexNet	-
Kiran et al., 2023 [129]	Radiography	UNet	-
Lee et al., 2022 [114]	PubMed and EMBASE databases	Autoencoder	-
Calivà et al., 2022 [126]	MRI	Deep neural networks	-
Smolle et al., 2023 [120]	X-ray	KL grade, joint space narrowing (JSN), sclerosis and osteophyte OARSI grade by computerized methods	-
Lenskjold et al., 2022 [121]	X-ray	KL and OARSI grades	-
Xu et al., 2023 [122]	CBCT	KNN and convolutional neural networks (CNN)	-
Roemer et al., 2024 [130]	MRI	Whole-Organ Magnetic Resonance Imaging Score (WORMS)	-
Neubauer et al., 2023 [123]	X-ray	Correlation analysis	-
Tiwari et al., 2022 [124]	X-ray	ResNet50, VGG-16, InceptionV3, MobilnetV2, EfficientnetB7, DenseNet201, Xception and NasNetMobile	-

## Data Availability

There were no associated data with this manuscript.

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
