# Peer review of "Minimally Invasive Therapies for Knee Osteoarthritis"

_jpm, 2024, doi:10.3390/jpm14090970_

Round 1

Reviewer 1 Report

Comments and Suggestions for Authors

1. This is a good review to demonstrate current Minimally Invasive Therapies for KOA treatment.

2. Information about Physical therapy, such as electro-frequency, photodynamic therapy, infrared heat therapy, and even acupuncture as traditional homeopathy etc. could as addition for Minimally Invasive Therapies for Knee Osteoarthritis.

3. A list to show all Current Management would be helpful for the reader.

4. Any limitation when we apply AI in KOA management?

5. Conclusion should include all Minimally Invasive Therapies for Knee Osteoarthritis, rather than only focus on AI Performance.

Author Response

Reviewer #1 Comment #1: Comments and Suggestions for Authors. 1. This is a good review to demonstrate current Minimally Invasive Therapies for KOA treatment.

RESPONSE: We thank the reviewer for their strong support and for their time reviewing our work.

Reviewer #1 Comment #2: 2. Information about Physical therapy, such as electro-frequency, photodynamic therapy, infrared heat therapy, and even acupuncture as traditional homeopathy etc. could as addition for Minimally Invasive Therapies for Knee Osteoarthritis.

RESPONSE: We appreciate the reviewer for this suggestion. We have added a sentence to our section “2.3 Current management and challenges in treatment” in page 5, lines 130-133 “Other modalities like electro-frequency, photodynamic therapy, infrared heat therapy, electro-frequency, photodynamic therapy, infrared heat therapy, and even acupuncture as traditional homeopathy are quite minimally invasive; however, their clinical efficacy is uncertain or quite limited.”

Reviewer #1 Comment #3: 3. A list to show all Current Management would be helpful for the reader.

RESPONSE: We thank the reviewer for this great suggestion. We have created a new table in organizing the current management options by level of invasiveness and general type in page 5.

Reviewer #1 Comment #4: 4. Any limitation when we apply AI in KOA management?

RESPONSE: We appreciate the reviewer for this excellent suggestion. We have included extensive discussion on the limitations in the new version (page 22 lines 840-856: “Applying AI in the management of KOA presents several limitations. One major challenge is the quality and quantity of data available for training AI models. AI systems rely heavily on large, high-quality datasets to make accurate predictions and recommendations. However, KOA datasets can be limited by factors such as small sample sizes, variability in imaging techniques, and inconsistencies in diagnostic criteria. These limitations can affect the generalizability and accuracy of AI models, potentially leading to less reliable predictions and interventions. Furthermore, the complexity of KOA, which includes diverse patient demographics and disease stages, can complicate the development of one-size-fits-all AI solutions. Another significant limitation is the interpretability and integration of AI findings into clinical practice. While AI can analyze complex patterns in data, the results are often presented as black-box predictions without clear explanations of the underlying reasoning. This lack of transparency can hinder clinicians' trust and ability to make informed decisions based on AI recommendations. In addition, integrating AI tools into existing healthcare systems can be challenging due to issues related to interoperability, data privacy, and the need for clinician training. These barriers can impede the effective implementation and utilization of AI in managing KOA, limiting its potential benefits in improving patient outcomes.”

Reviewer #1 Comment #5: 5. Conclusion should include all Minimally Invasive Therapies for Knee Osteoarthritis, rather than only focus on AI Performance.

RESPONSE: We thank the reviewer and agree with their point. We have created a new section for the conclusion that summarizes the entire paper without just a focus on AI. This can be found in page 23 under section “12. Conclusion”.

Reviewer 2 Report

Comments and Suggestions for Authors

This study provides an extensive review of the diagnosis of osteoarthritis and recent minimally invasive treatments. In addition to the latest findings on hyaluronic acid injections, steroid injections, and PRP, it is particularly interesting that the study investigates and lists recent innovative treatments, including knee artery embolization, knee nerve ablation, regenerative medicine, and the role of AI.

This paper can be considered highly beneficial for clinicians in acquiring recent knowledge.

As an additional suggestion, including the recommended levels of AAOS and other guidelines (OARSI, NICE, ACR, etc.) in the conclusions of each section would make it much clearer.

Author Response

Reviewer #2 Comment #1: This study provides an extensive review of the diagnosis of osteoarthritis and recent minimally invasive treatments. In addition to the latest findings on hyaluronic acid injections, steroid injections, and PRP, it is particularly interesting that the study investigates and lists recent innovative treatments, including knee artery embolization, knee nerve ablation, regenerative medicine, and the role of AI.

This paper can be considered highly beneficial for clinicians in acquiring recent knowledge.

As an additional suggestion, including the recommended levels of AAOS and other guidelines (OARSI, NICE, ACR, etc.) in the conclusions of each section would make it much clearer.

RESPONSE: We thank the reviewer for their strong support of our manuscript. After each treatment modality, we have a section dedicated to “Recommendations and Conclusions” that already mentions the recommendations from AAOS, in addition to AAFP. However, we have added several recommendations from other guidelines, like those suggested, throughout the manuscript of the new version.